

# A new method for ecoacoustics? Toward the extraction and evaluation of ecologically-meaningful soundscape components using sparse coding methods

Alice Eldridge[1], Michael Casey[2], Paola Moscoso[1] and Mika Peck[1]

[1] Department of Evolution, Behaviour and Environment, University of Sussex, Brighton, East Sussex, UK
[2] Departments of Music and Computer Science, Dartmouth College, Hanover, NH, United States

## ABSTRACT

Passive acoustic monitoring is emerging as a promising non-invasive proxy for ecological complexity with potential as a tool for remote assessment and monitoring (*Sueur & Farina*, *2015*). Rather than attempting to recognise species-specific calls, either manually or automatically, there is a growing interest in evaluating the global acoustic environment. Positioned within the conceptual framework of ecoacoustics, a growing number of indices have been proposed which aim to capture community-level dynamics by (e.g., *Pieretti, Farina & Morri*, *2011*; *Farina*, *2014*; *Sueur et al.*, *2008b*) by providing statistical summaries of the frequency or time domain signal. Although promising, the ecological relevance and efficacy as a monitoring tool of these indices is still unclear. In this paper we suggest that by virtue of operating in the time *or* frequency domain, existing indices are limited in their ability to access key structural information in the spectro-temporal domain. Alternative methods in which time-frequency dynamics are preserved are considered. Sparse-coding and source separation algorithms (specifically, shift-invariant probabilistic latent component analysis in 2D) are proposed as a means to access and summarise time-frequency dynamics which may be more ecologically-meaningful.

## INTRODUCTION

Biodiversity assessment is a central and urgent task, not only for research in the biological sciences, but also in applied conservation biology, including major multi-lateral initiatives for promoting and protecting biodiversity. At the governmental level biodiversity needs to be incorporated into national accounting by 2020 (Aichi Biodiversity targets A2) (http://www.cbd.int/sp/targets/) and cost effective tools necessary to achieve this remain elusive. Operating within the conceptual and methodological framework of the burgeoning field of ecoacoustics, (*Sueur & Farina*, *2015*) we are interested in the potential for investigating the acoustic environment–or soundscape–as a resource from which to

Corresponding author
Alice Eldridge, aliceе@sussex.ac.uk

infer ecological information. The main contribution of this paper is to highlight a disjunct between a founding premise of ecoacoustics (that the acoustic environment is structured through spectro-temporal partitioning) and the fact that community level indices to date are derived from representations of the acoustic signal in the time *or* frequency domain and therefore limited in accessing and evaluating structures across spectro-temporal dimensions. We consider approaches to decomposition which preserve time-frequency structure and propose sparse-coding as a possible solution. Ecoacoustic applications are illustrated with example analyses from a recent acoustic survey. Results are illustrative rather than conclusive but point to possibilities for analyses of community level acoustic structures which are impervious to investigation with current tools.

## Ecoacoustic approaches to biodiversity assessment

In ongoing work, we are exploring cost-effective solutions, including remote sensing (camera traps and aerial photography of canopy) and identification of 'ecological-disturbance indicator species' (*Caro*, *2010*). Remote sensors are an attractive choice for data collection in that they are noninvasive, scalable in both space and time and remove the bias and cost associated with programs which require either experts (All taxa biodiversity inventory, *Gewin*, *2002*) or even non-specialists (Rapid Biodiversity Assessment, *Oliver & Beattie*, *1993*), *in situ*.

Various forms of remote visual sensing technologies have been explored. Global satellite imaging has been investigated to monitor biophysical characteristics of the earth's surface by assessing species ranges and richness patterns indirectly (e.g., *Wang et al.*, *2010*). These methods are attractive, but rely on expensive equipment, are difficult to adapt to small spatial scales and require a time-consuming validation step. It is possible, for example, to infer valid species-level identification of canopy trees from high-resolution aerial imagery, providing a means of remote sensing to assess forest status (*Peck et al.*, *2012*). However, the principal weakness of this and other existing visual remote sensing methods is that they cannot provide direct information on the status of taxa other than plants: they cannot detect 'silent forests.' The need for innovative remote sensing methods to monitor the status of wildlife remains and acoustic, rather than visual, sensors have many attractive characteristics.

Acoustic surveys have most obvious relevance for the identification of vocal animals. In terrestrial habitats, bird species in particular are of interest as their importance as indicator species of environmental health has been demonstrated in temperate (*Gregory & Strien*, *2010*) and tropical (*Peck et al.*, *2015*) climates. One approach is to focus on automatic species call identification, but current methods are far from reliable (e.g., *Skowronski & Harris*, *2006*, for bats), increasingly difficult in complex environments such as tropical forest soundscapes, where tens of signals mix and many species still remain unknown (*Riede*, *1993*) and notoriously difficult to generalize across locations due to natural geographic variation in species' calls (*Towsey et al.*, *2013*).

Rather than focusing on individual species, there is a growing interest in monitoring high-level structure within the emerging field of *Soundscape Ecology* (*Pijanowski et al.*, *2011*) in which systematic interactions between animals, humans and their environment

are studied at the landscape level. From this emerging perspective, the landscape's acoustic signature–the *soundscape*–is seen as a unique component in the evaluation of its function, and therefore potential indicator of its status (*Krause, 1987*; *Schafer, 1977*). We can consider similar processes occurring at the community level: vocalising species establish an *acoustic community* when they sing at the same time at a particular place. The potential for estimation of acoustic community dynamics as key to understanding what drives change in community composition and species abundance is being recognised (*Lellouch et al., 2014*). The nascent discipline of ecoacoustics unites theoretical and practical research which aims to infer ecological information from the acoustic environment across levels (*Sueur & Farina, 2015*) and habitats. In this paper we focus on terrestrial habitats, although the discussion is equally applicable to aquatic environments.

The motivations of the ecoacoustic approach can be understood in evolutionary terms: the same competitive forces which drive organisms to partition and therefore structure dimensions of their shared biophysical environment (food supply, nesting locations etc.) apply in the shared sonic environment; the soundscape is seen as a finite resource in which organisms (including humans) compete for spectro-temporal space. These ideas were first explicitly captured in Krause's *Acoustic Niche Hypotheses* (ANH) (*Krause, 1987*). Referring directly to Hutchinson's original niche concept (*Hutchinson, 1957*) the ANH suggests that vocalising organisms have evolved to occupy unique spectro-temporal 'niches,' minimising competition and optimising intraspecific communication mechanisms. Formulated following countless hours recording in pristine habitats, Krause goes so far as to posit that this spectro-temporal partitioning structures the global soundscape, such that the global compositional structure is indicative of the 'health' of a habitat. Crudely put, in ancient, stable ecosystems, the soundscape is expected to comprise a complex of non-overlapping signals well dispersed across spectro-temporal niches; a newly devastated area might be characterised by gaps in the spectro-temporal structure; and an area of regrowth may comprise competing, overlapping signals due to invasive species.

Krause's ANH can be understood in terms of several theories of the evolution of bird species, which are supported by field studies. Avian mating signals are thought to diverge via several processes: (1) as a by-product of morphological adaptation, the *Morphological Adaptation Hypothesis*; (2) through direct adaptation to physical features of the signalling environment, the *Acoustic Adaptation Hypothesis*; and (3) to facilitate species recognition, the *Species Recognition Hypothesis*. Field studies of the Neotropical suboscine antbird (Thamnophilidae) provide direct evidence that species recognition and ecological adaptation operate in tandem, and that the interplay between these factors drives the evolution of mating signals in suboscine birds (*Seddon, 2005*). Although the ANH is challenged by various studies in terrestrial and marine environments (*Amézquita et al., 2011*; *Chek, Bogart & Lougheed, 2003*; *Tobias et al., 2014*), it is tenable in evolutionary terms and acoustic partitioning in both time and frequency domains have been observed (*Sinsch et al., 2012*; *Schmidt & Balakrishnan, 2015*; *Ruppé et al., 2015*).

## The constraints of existing acoustic indices for automated ecoacoustics

This emerging framework, coupled with the technical feasibility of remote acoustic sensing and pressure to meet strategic biodiversity targets, fuels a growing research interest in ecological applications of acoustic indices; several dozen have been proposed over the last 6 years (see *Sueur et al., 2014*; *Towsey et al., 2013*; *Lellouch et al., 2014*, for good overviews in terrestrial habitats and *Parks, Miksis-Olds & Denes, 2014*, for marine acoustic diversity). These are motivated by different approaches to measuring the variations in acoustic activity and predominantly derived from statistical summaries of amplitude variation in time domain *or* magnitude differences between frequency bands of a spectrogram.

The simplest indices provide summaries of the Sound Pressure Level (e.g., peaks, or specific times of day). In *Rodriguez et al. (2013)*, for example, root mean square values of raw signals from a network of recorders are used to create maps of amplitude variation to reveal spatiotemporal dynamics in a neotropical forest.

Under the assumption that anthropogenic noise contribution is band-limited to a frequency range (anthropophony: 0.2–2 kHz) below that of the rest of the biological world (biophony: 2–8 kHz), the **Normalized Difference Soundscape Index** (NDSI) (*Kasten et al., 2012*) seeks to describe the 'health' of the habitat in terms of the level of anthropogenic disturbance by calculating the ratio (biophony − anthropophony) / (biophony + anthropophony). In long term studies, the NDSI has been shown to reflect assumed seasonal and diurnal variation in a landscape and may prove useful for observing high level, long term interactions between animals and human populations (*Kasten et al., 2012*). However, it does not give an estimation of local diversity *within* the range of biophony, or provide a means to investigate short term interactions in detail. Further, assumptions about frequency ranges may not generalize. For example in non-industrialized tropical climes (arguably the most precious in ecological terms) animals vocalize outside the 2–8 kHz range (*Sueur et al., 2014*), and industrial anthropophony is minimal. Similarly, in marine habitats these frequency ranges may not be relevant.

A range of entropy indices are based on the assumption that the acoustic output of a community will increase in complexity with the number of vocalising individuals and species. A summary of the complexity of the sound is assumed to give a proxy of animal acoustic activity. Complexity here is used as a synonym of heterogeneity and many indices derive from classical ecological biodiversity indices. Shannon Entropy (*Shannon & Weaver, 1949*) Eq. 1 is favoured by ecologists as a measure of species diversity, where $p_i$ is the proportion of individuals belonging to the $i_{th}$ species in the data set of interest; it quantifies the uncertainty in predicting the species identity of an individual that is taken at random from the dataset.

$$H' = -\sum_{i=1}^{R} p_i ln p_i \tag{1}$$

The **Acoustic Entropy Index**, $H$ (*Sueur et al., 2008b*) is described as the product of spectral ($sh$) and temporal ($th$) entropies which are calculated on the mean spectrum

and Hilbert amplitude envelope of a time wave respectively. *H* ranges from 0 for pure tones to 1 for high-energy, evenly distributed sound. The index was first tested against simulated choruses, generated by mixing together samples of avian vocalisations and systematically varying the number of species in each track. H values increased with species richness S following a logarithmic model. Field trials were carried out in pristine and degraded African coastal forests and *H* was shown to reflect assumed variation in species richness (*Sueur et al.*, *2008b*). The study was in an area where animal acoustic activity was high and background noise low. When background noise (such as traffic) or broadband signals (such as rain, cicada or tropical cricket choruses) are higher, spectral entropy measures may give counter-intuitive results–times of low acoustic activity with relatively loud background noise for example would approach 1. This is an real issue in passive acoustic monitoring in both terrestrial and marine environments as the low sensitivity of rugged outdoor microphones tend to create a high background noise level.

The **Acoustic Diversity Index** (*Villanueva-Rivera et al.*, *2011*) (ADI) is a spectral entropy measure which summarises the distribution of the proportion of signals across the spectrum. The FFT spectrogram is divided into a number of bins (default 10) the proportion of the signals in each bin above a threshold (default = 50 dBFS) is calculated. The Shannon Index Eq. 1 is then applied, where $p_i$ is the fraction of sound in each $i_{th}$ of *R* frequency bands. An evenness metric, the **Acoustic Evenness Index** (AEI) is similarly derived by calculating the Gini index (*Gini*, *1912*) (commonly used by ecologists to estimate species evenness) on the spectrum. These relatively simple indices are shown to effectively reflect observed distinctions in gross acoustic activity, for example between dawn choruses and night activity, or between diverse habitats (mature oak forest, secondary forest, wetland and agricultural land).

The spectral indices provide a statistical summary of the distribution of energy across the sample, typically 1–10 mins are analysed at a time. These prove useful in long term studies or for observing gross changes in time or space. Seeking to capture subtler changes in behaviour and composition of vocalising communities, and to counter the noise-sensitivity of the entropy indices, the **Acoustic Complexity Index** (ACI) was developed specifically to capture the dynamic changes in the soundscape: "many biotic sounds, such as bird songs, are characterised by an intrinsic variability of intensities, while some types of human generated noise (such as car passing or airplane transit) present very constant intensity values" (*Pieretti, Farina & Morri*, *2011*). The ACI is derived from measures of absolute difference in adjacent bins in a spectrogram and was shown to correlate with the number of bird vocalisations in a small scale spatial study in an Apennine National Park, Italy (*Pieretti, Farina & Morri*, *2011*).

The **Bioacoustic Index** (*Boelman et al.*, *2007*) is presented as a measure of avian abundance and is calculated simply as the area under the mean frequency spectrum (minus the value of the lowest bin), providing a measure of both the sound level and the number of frequency bands used by the avifauna. It was used to investigate differences between exotic and native species in Hawaii and shown to be strongly correlated with counts from direct ornithological survey when calculated for single samples taken across a six week period.

Initial studies are encouraging: indices have been shown to correlate with aurally identified changes in bird species richness (*Depraetere et al.*, *2012*) and reveal dynamic variation across landscape, however there are many open questions both methodologically and theoretically. Existing indices are inherently likely to be affected by several factors including transitory or permanent background noise, variation in distance of the animal to the microphone and relative intensity of particular species call patterns. Theoretically, we are still far from understanding exactly what aspects of biodiversity these indices might represent (*Pijanowski et al.*, *2011*; *Sueur et al.*, *2008b*; *Servick*, *2014*). This is highlighted in a recent temporal study of dissimilarity indices (*Lellouch et al.*, *2014*) in which indices were shown to correlate well with diversity of simulated communities, but did not track community composition changes in the wild, raising the question of what, if any, aspect of compositional diversity such indices represent?

By virtue of being based on either time-averaged spectrograms *or* amplitude changes in the time domain, we argue here that such indices are fundamentally limited in their ability to detect patterns *across* the spectro-temporal domain, which may be key to evaluating the acoustic dynamics of specific communities. Frequency-based indices can pick up on crude differences in gross frequency range; time domain indices can pick up changes in amplitude; both are inherently constrained in their ability to detect global spectro-temporal patterns created by cohabiting species interacting in an acoustic community. As the motivational premise of the community level approach assumes that spectro-temporal partitioning is responsible for structuring the soundscape (in both marine and terrestrial habitats), this is a significant constraint. Although existing indices are very cheap in computational terms, the fear is that what is gained in computational efficiency is lost of ecological efficacy. Rather than collapsing the signal into one or other domain, we propose that the theoretical and practical strands of ecoacoustic research could be advanced by developing tools in which time-frequency structure is preserved.

## Sparse coding and latent component analysis

Time-frequency tradeoffs are an important issue in all signal processing tasks. Sparse coding is gaining popularity in brain imaging, image analysis and audio classification tasks as an alternative to vector-based feature representations in part because it is seen to have more time-frequency flexibility than standard Fourier transform representations. Sparse coding aims to construction efficient representations of data as a combination of a few typical patterns (atoms) learned from the data itself. For a given set of input signals, a number of atoms are sought, such that each input signal can be approximated sparsely by a linear combination of a relatively small number of this set of atoms (the dictionary). The dictionary size is higher than the dimensionality of the signal such that a subset of atoms can span the whole signal space—an overcomplete dictionary (*Scholler & Purwins*, *2011*). Sparse approximations of the signal are then constructed by finding the "best matching" projections of multidimensional data onto an over-complete dictionary, Matching Pursuit (*Mallat & Zhang*, *1993*) (MP) being a popular choice.

Sparse decomposition using dictionaries of atoms based on biologically informed time-frequency atoms such as Gabor and Gammatone functions–which are seen to

resemble characteristics of cochlea filters–are intuitively attractive as they can provide a feature set which is oriented in a two dimensional time-frequency space with which to approximate the original signal. This has been shown to be more efficient than Fourier or wavelet representations (*Smith & Lewicki*, *2005*) and to provide effective and efficient input features in a range of audio discrimination tasks in everyday sounds (*Adiloglu et al.*, *2012*), drum samples (*Scholler & Purwins*, *2011*) and similarity matching of bioacoustic data (*Glotin et al.*, *2013*).

Probabilistic Latent Component Analysis (PLCA) is one of a family of techniques used for source separation, which similarly provides a tool for extracting components according to common frequency-amplitude statistics. PLCA is a probabilistic variant of non-negative matrix factorization (NMF) (*Lee & Seung*, *2001*). It decomposes a non-negative matrix $V$ into the product of two multinomial probability distributions, $W$ and $H$, and a mixing weight, $Z$. In the auditory domain, V would be a matrix representing the time-frequency content of an audio signal:

$$V \approx WZH = \sum_{k=0}^{K-1} w_k z_k h_k^T \qquad (2)$$

where each column of $W$ can be thought of as a recurrent frequency template and each row of $H$ as the excitations in time of the corresponding basis. $Z = \mathrm{diag}(z)$ is a diagonal matrix of mixing weights $z$ and $K$ is the number of bases in $W$ (*Weiss & Bello*, *2010*). Each of $V$, $w_k$, $z_k$, and $h_k$ correspond to probability distributions and are normalized to sum to 1.

PLCA can be compared to more familiar component analysis tools such as Principle or Independent Component Analyses (PCA, ICA) and can be used to perform dimensionality reduction, feature extraction or to explore structure in a data set. The non-negativity constraint is a valuable property for audio and image decompositions, where non-negative representations are prevalent, as the non-negative elements are often perceptually meaningful decompositions which can be easily interpreted. By comparison, methods using non-negativity are bound to return bases that contain negative elements and then employ cross-cancellation between them in order to approximate output (*Smaragdis, Raj & Shashanka*, *2008*). Such components are harder to interpret in a positive only setting and although useful for their statistical properties provide little insight.

Sparse and shift-invariant PLCA (SI-PLCA) extends PLCA to enable the extraction of multiple shift-invariant features from analysis of non-negative data of arbitrary dimensionality and was first demonstrated as an effective unsupervised tool for extracting shift-invariant features in images, audio and video (*Smaragdis, Raj & Shashanka*, *2008*). The algorithm provides a very precise and perceptually meaningful description of content. A series of piano notes, for example, is automatically decomposed into a kernel distribution representing the harmonic series common to all notes, the peaks of the impulse distribution representing the fundamental frequency of each note and its location in time (*Smaragdis, Raj & Shashanka*, *2008*). *Weiss & Bello* (*2010*) demonstrated application in segmentation task, showing SI-PLCA to be competitive with Hidden Markov Models and self-similarity

matrices. More recently, *Sarroff & Casey* (*2013*) developed a shift and time-scale invariant PLCA which performed well against subjective human ratings of musical 'groove'—a multi-dimensional rhythmic musical feature correlated with the induction of bodily movement.

A common strategy used throughout the NMF literature is to favour sparse settings in order to learn parsimonious, parts-based decompositions of the data. Sparse solutions can be encouraged when estimating the parameters of the convolution matrix by imposing constraints using an appropriate prior distribution (*Smaragdis, Raj & Shashanka*, *2008*). Under the Dirichlet distribution for example, hyper-parameters can be set to favour a sparse distribution. In these cases, the algorithm will attempt to use as few bases as possible, providing an 'automatic relevance determination strategy' (*Weiss & Bello*, *2010*): the algorithm can be initialised to use many bases; the sparse prior then prunes out those that do not contribute significantly to the reconstruction of the original signal. In the context of pop song segmentation, this enables the algorithm to automatically learn the number and length of repeated patterns in a song. In soundscape analysis, might this provide an ecologically-relevant indicator of the compositional complexity of an acoustic community?

In Music Information Retrieval (MIR) and composition tasks, SI-PLCA provides a tool for accessing perceptually meaningful decompositions—time-frequency shifted patterns in a dynamic signal. From the perspective of community level ecoacoustics we are not necessarily concerned with the identification of specific species, so much as achieving a numerical description of the *qualitative* patterns of interaction between them. By way of musical analogy, we don't care what the specific instruments of the orchestra are, rather we wish to assess characteristics of the arrangement and how the voices interact as an ensemble toward a coherent global composition through time, timbre and frequency space. Frequency-based indices may succeed in tracking species richness in simulated communities by measuring gross changes in frequency band occupancy. Perhaps their failure to track variation in species richness in the wild is because the defining feature of acoustic communities are global patterns of interaction across a more complex *spectro-temporal* domain, rather than frequency band occupancy or amplitude variation alone. As outlined above, current indices based on frequency *or* amplitude statistics inherently throw away information crucial to the analysis of spectro-temporal patterns: SI-PLCA provides a tool for extracting dynamic sound components grouped by common frequency-amplitude statistics, even when pitch or time shifted. That it has been demonstrated to be effective in extracting the perceptually-meaningful but nebulous concept of 'groove' (*Sarroff & Casey*, *2013*) suggests potential as a tool for beginning to interrogate the multi-dimensional dynamic complexities of acoustic communities.

In this paper we take a first look at how these methods might provide a complementary approach to current acoustic indices for investigation of soundscape dynamics and ultimately for biodiversity assessment. Taking a small sample of field recordings across different terrestrial habitats in an Ecuadorian cloud forest reserve we compare existing spectral and temporal indices with sample analyses of a number of approaches to sparse approximation, including dictionaries built using mini-batch gradient descent, Gabor
functions and SI-PLCA2D. The potential value of this approach is illustrated with example reconstructions from a new variant of SI-PLCA using dual dictionaries.

## METHODS AND MATERIALS

### Data collection

#### Study area and acoustic survey methods

The data reported here is a subset of that collected during an 8 week field survey (June–August 2014) in the Ecuadorian Andean cloud forest at the Santa Lucia Cloud Forest Reserve (SLR)[1]. The SLR (0°07′30″N, 78°40′3″W) is situated on the western (Pacific) slopes of the Andes in northwestern Ecuador and spans an elevational range of 1,400–2,560 m. The forest is lower montane rain forest (cloud forest). The area has a humid subtropical climate and is composed of fragmented forest reserves surrounded by a matrix of cultivation and pasture lands. It lies within the Tropical Andes biodiversity hotspot and exhibits high plant species endemism and diversity. Topography is defined by steep-sloping valley systems of varying aspect.

The SLR was awarded reserve status 20 years ago, prior to which areas of Primary Forest had cleared for fruit farming. The SLR therefore consists of a complex mosaic of habitat types: Ancient Primary Forest (FP) punctuated by small areas of secondary regrowth of around 20 years (FS) and silvopasture (S), typically elephant grass pastures used as grazing paddocks for the mules, which provide local transport. These areas are less than 5 ha. In contrast to other studies where dramatically different sites have been used to validate indices, this complex patchy habitat provides subtle habitat gradients.

Acoustic data was collected using nine digital audio field recorders Song Meter SM2+' (Wildlife Acoustics), giving three replicates of each of the three habitat types. Minimum distance between recorders was 300 m to avoid pseudo sampling (the sound of most species being attenuated over this distance in cloud forest conditions). Altitudinal range was minimised. Recording schedules captured the full dawn (150 min), dusk choruses (90 min) plus midday (60 min) activity; throughout the rest of the period 3 min recordings were taken every 15 min and ran for a minimum of 14 days at each study site.

The SM2+ is a schedulable, off-line, weatherproof recorder, with two channels of omni-directional microphone (flat frequency response between 20 Hz and 20 kHz). Gains were set experimentally at 36 dB and recordings made at 16 bit with a sampling rate of 44.1 kHz. All recordings were pre-processed with a high pass filter at 500 Hz (12 dB) to attenuate the impact of the occasional aircraft and local generator noise.

#### Species identification

A local expert ornithologist carried out point count surveys (*Ralph, Sauer & Droege*, *1995*), noting all birds seen and heard at each survey point for 10 min periods during the dawn chorus. A record was made for each individual, rather than individual vocalisations, and distance estimates given, providing species presence–absence and abundance measures.

### Acoustic indices

For the purposes of this illustrative exercise, analyses were carried out on dawn chorus recordings from just one day at one recording station for each of the three

[1] Field work data collection was authorised by the Ministerio del Ambiente, Ecuador (permit No 008–15 -IC-FAU-DNB/MA)

habitat types sampled. A range of indices frequency and time domain indices were calculated: NDSI, H, ADI, AEI, ACI and BI. Indices were calculated for the same 10 min periods during which point counts were made at each site. Calculations were made using the seewave (*Sueur, Aubin & Simonis*, *2008a*; Available at: http://cran.r-project.org/web/packages/seewave/index.html) and soundecology (Available at: http://cran.r-project.org/web/packages/soundecology/index.html) packages in R.

### Audio spectrum approximation methods

Three approaches to audio decomposition are illustrated using the Bregman Media Labs Audio Patch Approximation Python package (https://github.com/bregmanstudio/audiospectrumpatchapproximation): dictionary learning using mini-batch gradient learning, a Gabor field dictionary and, shift invariant 2D Principle Latent Component Analysis (SI-PLCA2D). Each uses Orthogonal Matching Pursuit (OMP) to build the component reconstructions. Samples were extracted from analyses of 1 min sections of the field recordings. These examples are aimed at illustrating the potential of a two-dimensional (2D) atomic rather than 1D vector approach in general, rather than experimental validation of any particular algorithm. Default parameters were used in all cases.

A potential future direction is illustrated using a SI-PLCA variant (SI-PLCA2) using 2D dual dictionaries (*Smaragdis & Raj*, *2007*; *Weiss & Bello*, *2010*; *Sarroff & Casey*, *2013*) based on frequency * local time functions and frequency-shift * global time-activations. The expectation-maximisation (EM) algorithm (*Smaragdis & Raj*, *2007*) is used to build component reconstructions.

As described in 'Sparse Coding and Latent Component Analysis', the algorithm returns a set of $k$ from a $K$ maximum components ($K_{max} = 16$): independent component reconstructions, time-frequency kernels and shift-time activation functions. Entropies of each are also returned. These example analyses are used to illustrate SI-PLCA as a potentially rich tool for future research in investigating the complex quasi-periodic signals of wild soundscapes.

## RESULTS AND DISCUSSION

### Species composition of acoustic communities

The species observations for each site, shown in Table 1, reveal little variation in overall abundance or species number between the sites when seen and heard records are considered together. Several species are observed in all sites; others are observed only in one habitat type. Discounting the seen-only counts, the highest number of species, and individuals, was recorded at S, with least heard at FP. The spectrograms and mean spectrum profiles (Fig. 1) for these recordings suggest that this information is present in the soundscape. The number of shared species between sites results in acoustic communities with an overall similar overlap, differentiated by calls of 'keynote' species. Each acoustic community occupies a broadly similar frequency range, with variation in the peaks of spectral profiles according to the prevalence of calls of habitat-specific species. FP appears to have lowest over-all activity, in line with the relatively fewer number of species observed.

**Table 1** **Field observations for species heard and seen.** Data shown for FP (0600-0610), FS (0619-0629) and S(0639-0649) on June 15th 2014.

| Common name | Heard | | | Seen | | |
|---|---|---|---|---|---|---|
| | FP | FS | S | FP | FS | S |
| Andean Solitaire | 1 | 1 | 1 | 1 | – | – |
| Azara's Spinetail | – | – | 1 | – | – | – |
| Beryl-spangled Tanager | – | – | – | 1 | 1 | – |
| Blue-winged Mountain-Tanager | – | 1 | – | – | – | 1 |
| Booted Racket-tail | – | 1 | – | 1 | – | – |
| Brown Inca | 1 | – | – | – | – | – |
| Brown-capped Vireo | 1 | – | – | – | – | – |
| Collared Forest-Falcon | 1 | – | – | – | – | – |
| Dusky Bush-Tanager | – | – | 1 | 1 | – | – |
| Empress Brilliant | – | – | 1 | – | – | 1 |
| Flame-faced Tanager | – | – | – | 1 | 1 | – |
| Golden-Crowned Flycatcher | – | – | 1 | – | – | – |
| Golden-Headed Quetzal | – | 1 | 1 | – | – | – |
| Gray-breasted Wood-Wren | 1 | 1 | 1 | – | – | – |
| Immaculate Antbird | – | 1 | – | – | – | – |
| Lineated Foliage-Gleaner | – | 1 | 1 | 1 | – | – |
| Long-tailed Antbird | 1 | – | 1 | – | – | – |
| Masked Trogon | 1 | 1 | 1 | – | – | – |
| Metallic-Green Tanager | – | – | 1 | – | – | – |
| Nariño Tapaculo | – | 1 | – | – | – | – |
| Orange-bellied Euphonia | 1 | 1 | – | 1 | – | – |
| Plumbeous Pigeon | 1 | 1 | 1 | – | – | 1 |
| Red-faced Spinetail | – | 1 | 1 | – | – | – |
| Roadside Hawk | – | – | 1 | – | – | – |
| Rufous-breasted Antthrush | – | 1 | – | – | – | – |
| Russet-crowned Warbler | 1 | 1 | – | – | – | – |
| Scale-crested Pygmy-Tyrant | – | 1 | 1 | – | – | – |
| Slate-throated Whitestart | 1 | – | 1 | – | – | – |
| Smoke-colored Pewee | – | – | 1 | – | – | – |
| Three-striped Warbler | – | 1 | – | – | – | – |
| Toucan Barbet | - | – | 1 | – | – | – |
| Tricolored Brush-Finch | – | 1 | 1 | – | – | – |
| Tyrannine Woodcreeper | – | – | – | 1 | – | – |
| Uniform Antshrike | – | – | 1 | – | – | – |
| Wattled Guan | 1 | 1 | 1 | – | – | – |
| White-capped Parrot | 1 | – | – | – | – | – |
| **Total** | 13 | 18 | 21 | 8 | 2 | 3 |

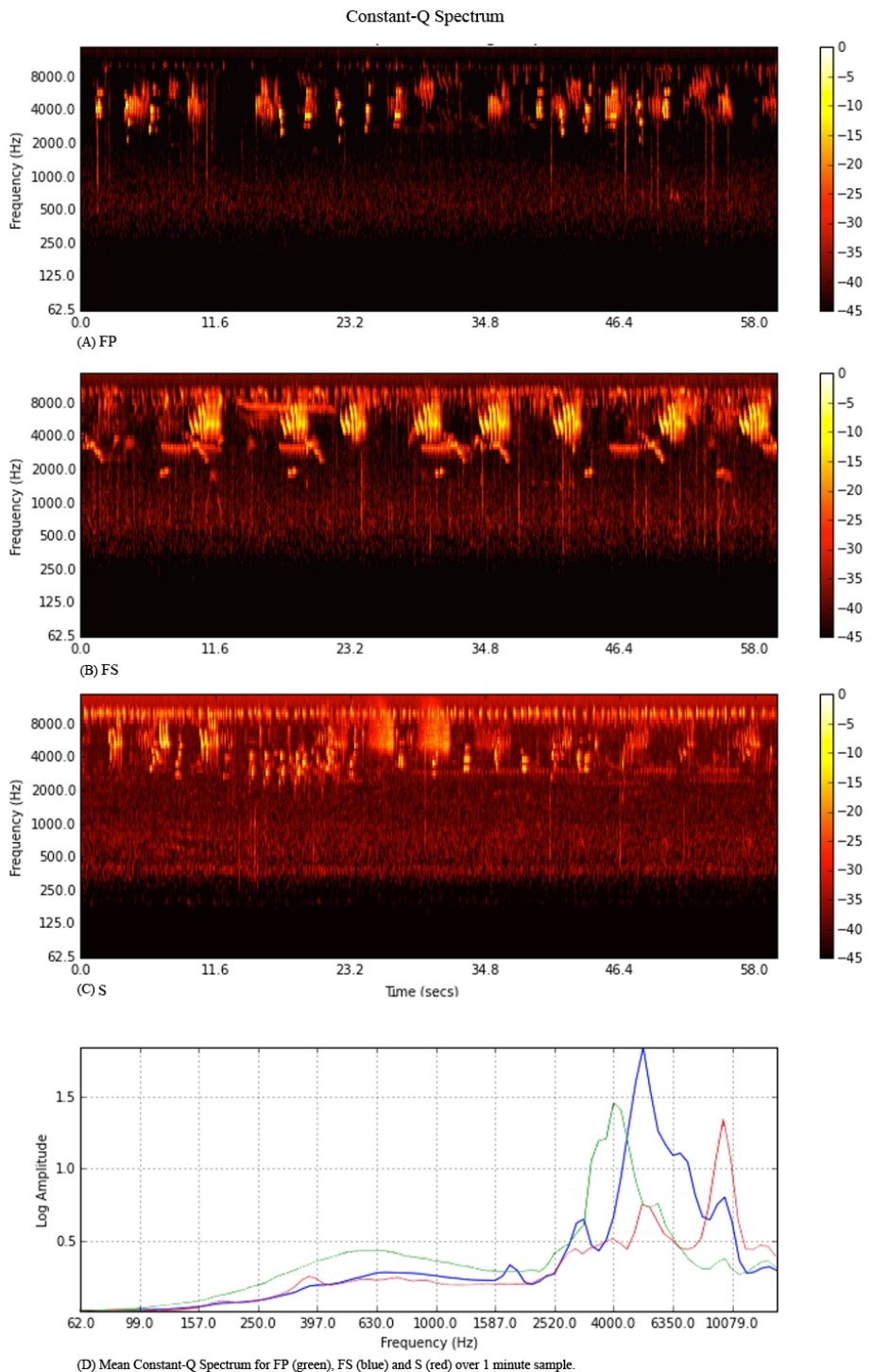

(D) Mean Constant-Q Spectrum for FP (green), FS (blue) and S (red) over 1 minute sample.

**Figure 1** **Constant-Q spectrograms for Primary (A) Secondary (B) and Silvopasture (C) sites.** 1 min resolutions are presented to illustrate the periodic call patterns. These were consistent across the 10 min sampling time in each habitat. Mean Constant-Q Spectrum (D) for Primary (green), Secondary (blue) and Silvopasture (red).

**Table 2**  Acoustic indices values for the three study sites: FP, FS and S.

| | NDSI | ADI | AEI | sh | th | H | ACI | BI |
|---|---|---|---|---|---|---|---|---|
| FP | 0.9716 | 2.1919 | 0.2591 | 0.9567 | 0.9730 | 0.9309 | 18497.14 | 7.5393 |
| FS | 0.9727 | 2.2684 | 0.1418 | 0.9355 | 0.9729 | 0.9102 | 18315.61 | 11.1780 |
| S | 0.9809 | 2.2909 | 0.0749 | 0.9539 | 0.9825 | 0.9372 | 18686.78 | 6.5867 |

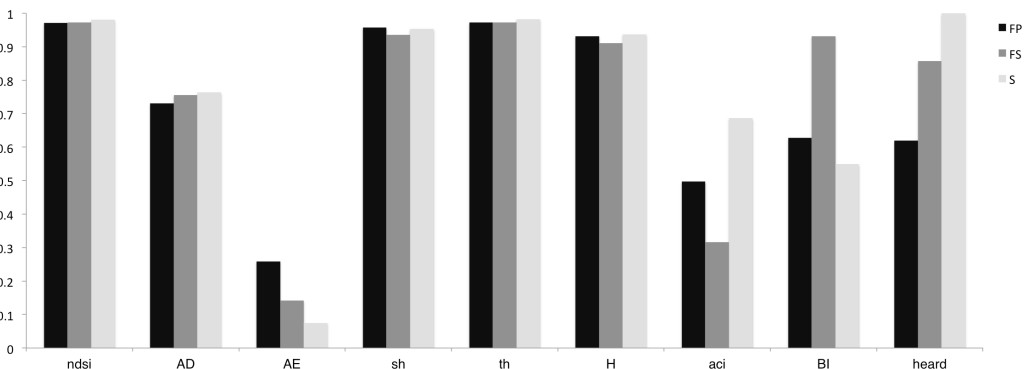

**Figure 2**  **Bar plots of indices values for the three study sites.** Plots show indices values for 10 min of dawn chorus. Values for AD, ACI and BI are scaled in the ranges 0:3, 1800:1900 and 0:12 respectively. Point count 'heard' data values for each site are given in the end column for comparison.

Despite occupying an overall similar frequency range and not differing dramatically in abundance, each site is distinctly characterised by differing quasi-periodic patterns of calls. The same patterns observed at the 1 min shown continued for the full 10 min sample[2]. The soundscape is structured, not just by repetitions of specific species calls, but by turn taking, i.e., interactions *between* species. This is most evident in listening, and can be observed visually as an interplay of periodic gestures in the spectrogram. It is precisely this complex of interacting periodic structures which we wish to evaluate under the soundscape approach, but which are impervious to analyses by current indices.

## Acoustic indices

Values for each of the acoustic indices calculated for the three habitats are given in Table 2 and shown as bar plots in Fig. 2. As we might expect given the minimal anthropogenic noise and broadly similar spectral profile, the NDSI reports near maximum values for each site. The global complexity of each scene is high; it is no surprise then that entropy indices approach 1 and differences between sites are minimal. The ADI reports a small variation, following the rank-order pattern of species heard at each site. Differences between Sueur's spectral, temporal and therefore overall, H entropy are minimal. ACI similarly shows small variation between sites. This index in particular is very sensitive to the size of the analysis window and requires further exploration to establish which aspects of community composition may be being assessed. BI values report the differences in overall acoustic energy, observable in mean spectrum plot (Fig. 1D), with the highest value at FS, FP being slightly higher than site S. These basic features of the acoustic recordings are at

[2] 1 min excerpts available in the Supplemental Information.

odds with the field observations of abundance and species numbers. An increase in overall energy could be due to certain individuals having intrinsically louder calls, calling more frequently, or simply being closer to the microphone. In validation studies the latter could be countered by factoring in field-based point count distance measures (recorded, but not included here) and call frequencies, as well as tallies of individual vocalisations, the latter being expedited by the use of automatic segmentation software (as in *Pieretti, Farina & Morri*, *2011*).

The key issue raised here, however, is that in providing summaries of frequency *or* temporal amplitude profile and magnitude differences, these current indices are not only sensitive to these largely irrelevant variations in overall amplitude changes, but are all *insensitive* to the periodic structures which uniquely characterise the three soundscapes.

## Sparse-approximation outputs

Dictionaries and sparse-approximations for FP using mini-batch gradient descent, Gabor atoms and SI-PLCA2D are shown in Fig. 3. The input for each is a log-frequency spectrogram (constant-Q transform) of samples from the field recordings, as shown in Fig. 1. Example dictionaries (Figs. 3A, 3C, 3E) and sparse approximations of the input spectrogram (Figs. 3B, 3D, 3F) for site FP are shown for each method (component reconstructions not shown). Comparing the sparse-approximation of the original spectrogram for FP (see Fig. 1A), the superior performance of SI-PLCA2D over the other two methods is evident.

The Gabor field dictionary has an intuitive advantage over vector descriptors in representing features oriented in time-frequency space. The dictionary learned under mini-batch gradient descent similarly exhibits time-frequency atoms differing subtly in orientation. The SI-PLCA2D dictionary however, comprises a collection of spectrum patches with a variety of micro-structures across a range of orientation and spread. In terms of the filter model which motivates the use of Gabor atoms, the Gabor and mini-batch dictionaries could be described as having relatively homogenous widths across the dictionary; the SI-PLCA2D dictionary by contrast contains points not only differing in time-frequency orientation, but in spectral width, atoms 0, 2, 3 and 4 being considerably more focused than 1 and 5 (Fig. 3E). This is an appealing property for the analysis of broad-spectrum versus pitched soundscape elements.

Full outputs for all three sites using the SI-PLCA2 algorithm with dual 2D dictionaries are shown in Figs. 4, 5 and 6. Each 10 min site recording is sampled, taking 16 time windows from across the file of around 4 s each, arranged in order. The input is the log-frequency spectrogram of these samples, as before. Extensive analysis of larger data sets across more diverse soundscapes is needed before we can begin to evaluate the ecological significance or application of this approach, but a number of promising observations can be made. As can be seen in Figs. 4A, 5A, 6A, the component reconstructions appear faithful to the original spectrogram. The individual component reconstructions (Figs. 4C, 5C, 6C) pull out clearly distinct components. This is clearest in S3 (Fig. 6C) where the first component is broadband ambient noise, and each of components 1–5 appear as distinct 'voices' grouped according to both spectral range and spectro-temporal periodic gesture.

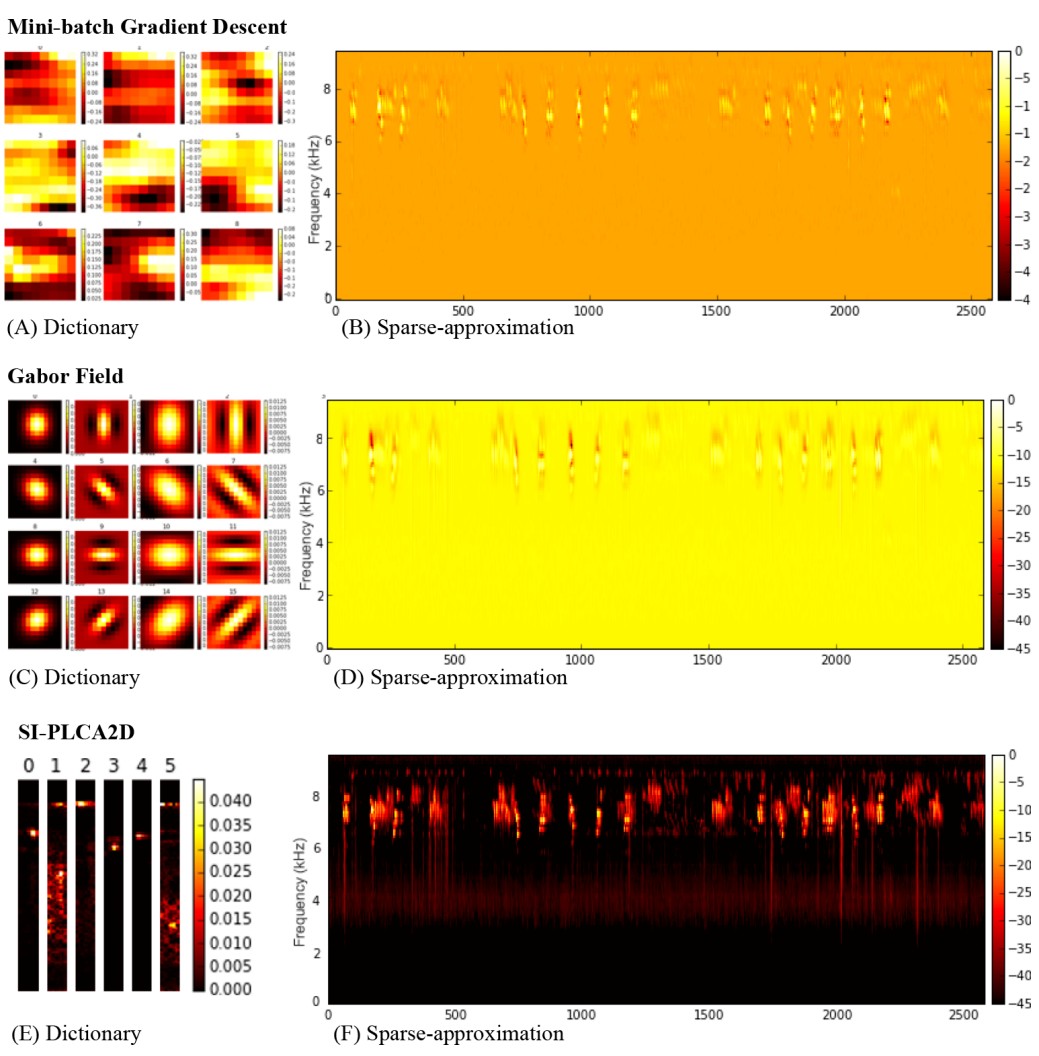

**Figure 3** Over-complete Dictionaries (A), (C), (E) and sparse-approximations of original spectrogram (B), (D), (F) for Primary Forest site dawn chorus for Mini-batch gradient learning, Gabor Field Dictionary and SI- PLCA2D Component Dictionary.

The time-frequency kernels provide a lower dimension representation of components with apparently similar characteristics: compare each component in Figs. 4C and 4D, for example. The automatic relevance determination feature deserves further investigation as a quick and dirty proxy for community composition assessment. In this example in FP k = 4, FS k = 7 and S k = 6. Does K increase with the number of vocalising species? Might it reflect the complexity or 'decomposability' of a scene in some way?

The entropies of each distribution are given in the subfigure captions of Figs. 4, 5 and 6. Whether these can provide useful information as a difference measure either between components *within* a particular reconstruction, or *between* kernels extracted from different soundscapes deserves further investigation. No conclusions can be drawn from this illustrative analysis, but it raises a number of questions for future research: (1) Are the component reconstructions meaningful soundscape objects in ecological

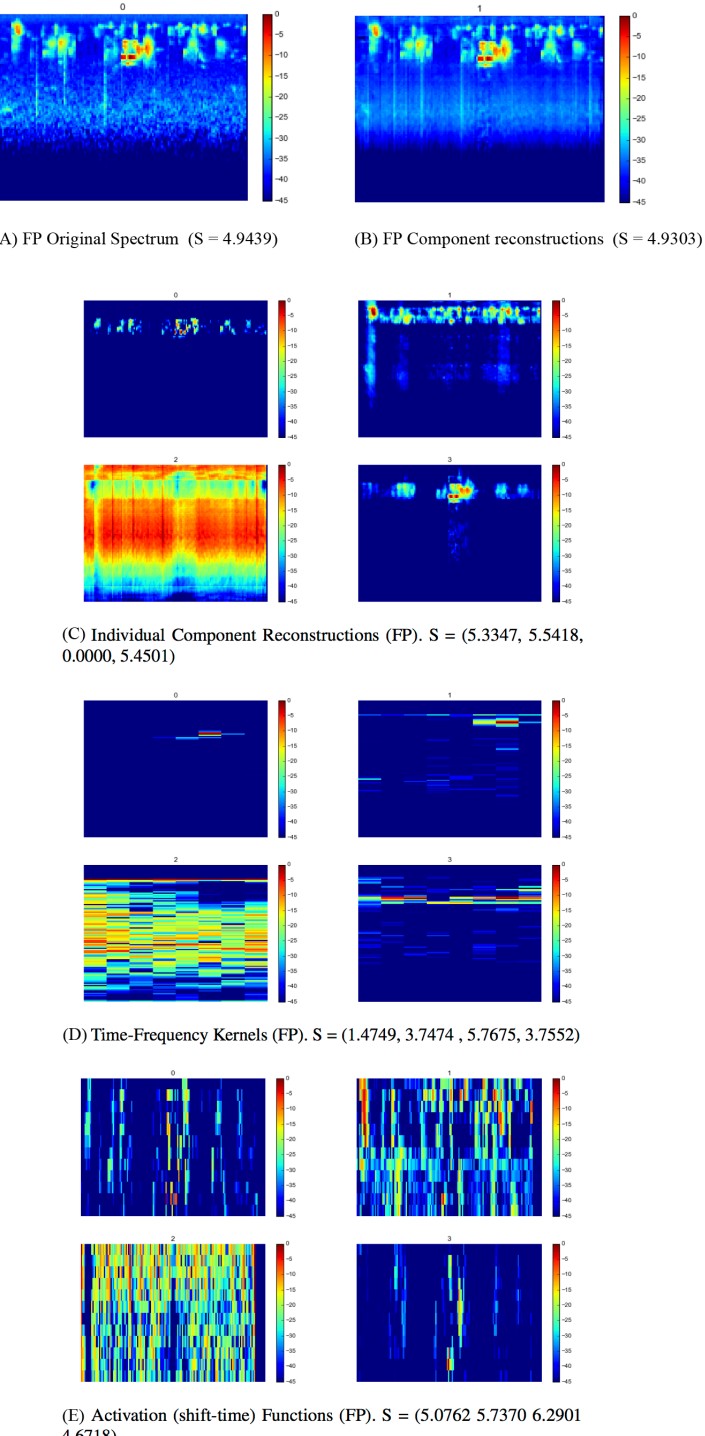

(A) FP Original Spectrum (S = 4.9439)  (B) FP Component reconstructions (S = 4.9303)

(C) Individual Component Reconstructions (FP). S = (5.3347, 5.5418, 0.0000, 5.4501)

(D) Time-Frequency Kernels (FP). S = (1.4749, 3.7474 , 5.7675, 3.7552)

(E) Activation (shift-time) Functions (FP). S = (5.0762 5.7370 6.2901 4.6718)

**Figure 4   SIPLCA2 outputs for Primary Forest site dawn chorus.** Entropy (S) values are shown in brackets. Original Spectrum (A) and Component Reconstructions (B), Individual Component Reconstructions (C), Time-Frequency Kernels (D) and Activation (shift-time) Functions (E).

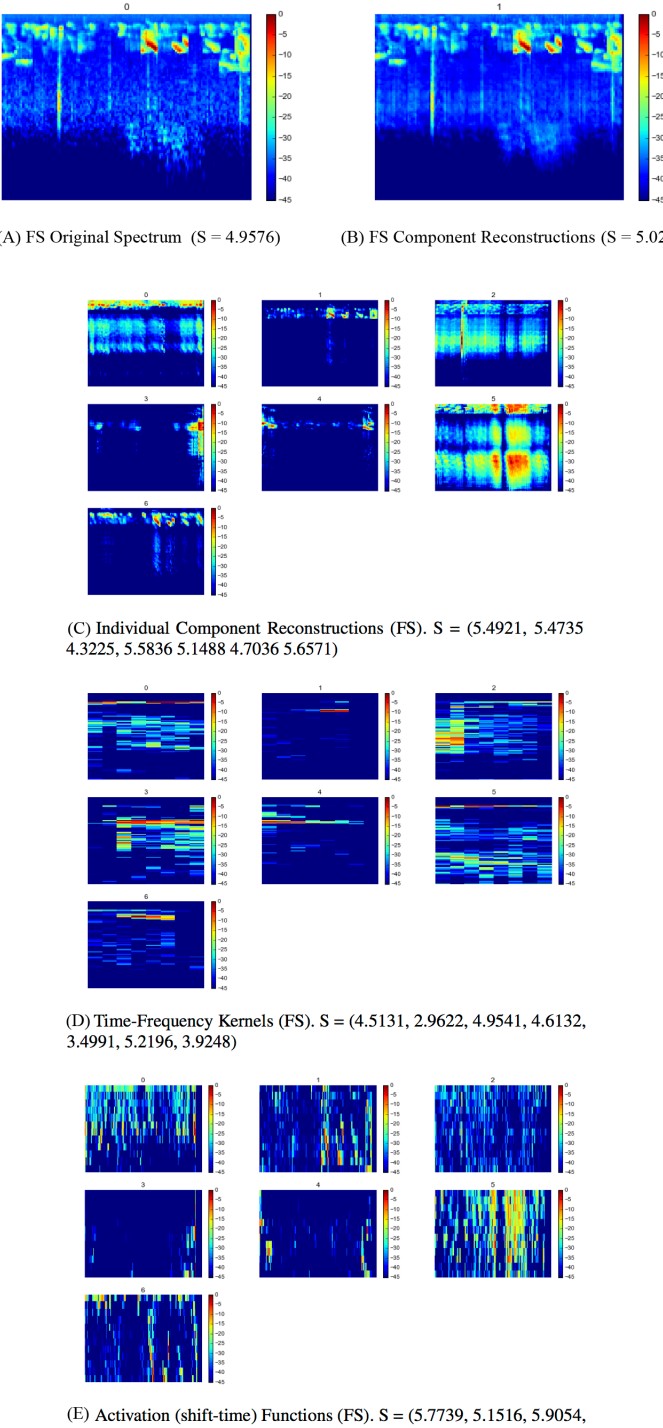

(A) FS Original Spectrum (S = 4.9576)    (B) FS Component Reconstructions (S = 5.0291)

(C) Individual Component Reconstructions (FS). S = (5.4921, 5.4735 4.3225, 5.5836 5.1488 4.7036 5.6571)

(D) Time-Frequency Kernels (FS). S = (4.5131, 2.9622, 4.9541, 4.6132, 3.4991, 5.2196, 3.9248)

(E) Activation (shift-time) Functions (FS). S = (5.7739, 5.1516, 5.9054, 3.6572, 4.2803, 6.0051, 4.9906)

**Figure 5  SIPLCA2 outputs for Secondary Forest site dawn chorus.** Entropy (S) values are shown in brackets. Original Spectrum (A) and Component Reconstructions (B), Individual Component Reconstructions (C), Time-Frequency Kernels (D) and Activation (shift-time) Functions (E).

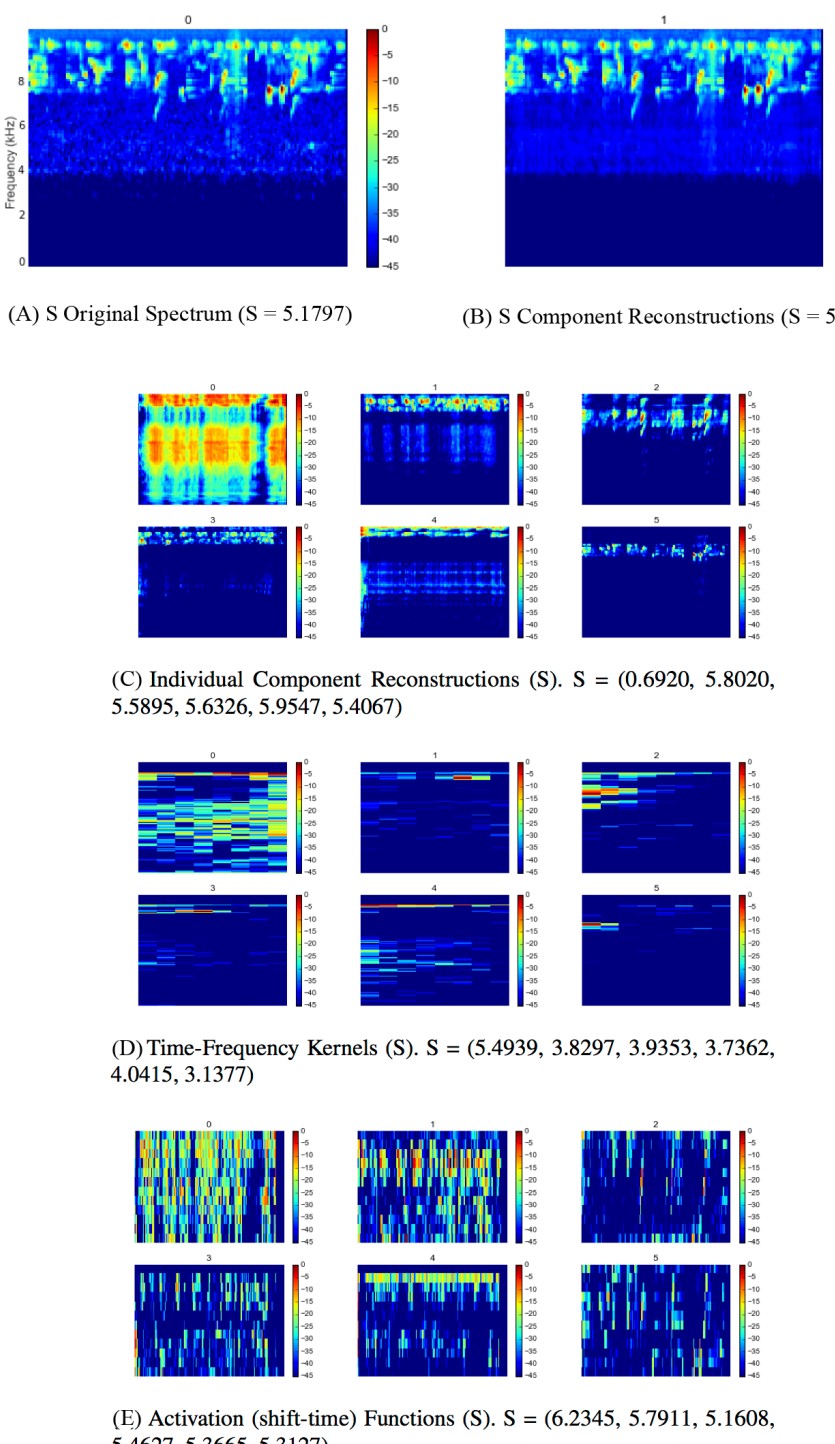

(A) S Original Spectrum (S = 5.1797)

(B) S Component Reconstructions (S = 5.1715)

(C) Individual Component Reconstructions (S). S = (0.6920, 5.8020, 5.5895, 5.6326, 5.9547, 5.4067)

(D) Time-Frequency Kernels (S). S = (5.4939, 3.8297, 3.9353, 3.7362, 4.0415, 3.1377)

(E) Activation (shift-time) Functions (S). S = (6.2345, 5.7911, 5.1608, 5.4627, 5.3665, 5.3127)

**Figure 6   SIPLCA2 outputs for Silvopasture site dawn chorus.** Entropy (S) values are shown in brackets. Original Spectrum (A) and Component Reconstructions (B), Individual Component Reconstructions (C), Time-Frequency Kernels (D) and Activation (shift-time) Functions (E).

terms? Are vocalising species separated in any meaningful ecological way either by soundscape component (geophony, biophony, anthrophony) or acoustic community (species, functionality etc.)? (2) Might the statistics generated be meaningful? Does the number of components ($k$) returned reflect 'complexity' or 'decomposability' in a way which may reflect the status of the acoustic community? Could the entropy summaries of each component be used as a measure of diversity within or between communities?

The ability of PLCA to separate streams of distinct sonic components is well recognised. Within the conceptual framework of ecoacoustics, such techniques provide a means to investigate the composition of the acoustic community as a whole in terms of dynamic interactions between spectro-temporal patterns of vocalising component species, providing a new tool to begin to experimentally interrogate the concept of *acoustic niche* in order to develop the understanding necessary to create more ecologically meaningful monitoring tools.

## SUMMARY AND FUTURE WORK

Monitoring subtle changes in complex ecosystems is crucial for ecological research and conservation but far from straight forward. Acoustic indices hold promise as a rapid assessment tool, but are subject to the same trade-offs as traditional ecological research of quality versus quantity: any metric necessarily throws away some information. In this paper we have provided an overview of the motivational premises of community-level ecoacoustics, including the concept that acoustic communities may be structured according to competition across acoustic niches through spectro-temporal partitioning. We suggest that existing indices operating in time *or* frequency domain may be insensitive to the dynamic patterns of interaction in the soundscape which characterise specific acoustic communities and propose SI-PLCA2D as a promising new tool for research. This was illustrated with example analyses of tropical dawn chorus recordings along a gradient of habitat degradation. It seems likely that if acoustic niches exist that they do not lie neatly along 1D vectors in the frequency or time domain but dance dynamically across pitch-timbre-time space. SI-PLCA2D and related sparse-coding methods are computationally expensive and do not offer an instant ready-to-use proxy for biodiversity monitoring. What they do provide is a tool for extracting shift-invariant spectro-temporal patterns in a dynamic soundscape, structures which are impervious to analysis with current tools. In future work we are testing the approach on more extensive data sets to establish the ecological meaning of the extracted components.

## ACKNOWLEDGEMENTS

Many thanks to Noé Morales of Santa Lucia for carrying out point count surveys.

### Funding

The research was funded by Leverhulme Trust Research Project Grant RPG-2014-403. The data collection was funded by University of Sussex Research Development Fund. The

funders had no role in study design, data collection and analysis, decision to publish, or preparation of the manuscript.

## Grant Disclosures
The following grant information was disclosed by the authors:
Leverhulme Trust Research Project: RPG-2014-403.

## Competing Interests
The authors declare there are no competing interests.

## Author Contributions
- Alice Eldridge conceived and designed the experiments, performed the experiments, analyzed the data, wrote the paper, prepared figures and/or tables, reviewed drafts of the paper, data Collection.
- Michael Casey conceived and designed the experiments, performed the experiments, analyzed the data, contributed reagents/materials/analysis tools, prepared figures and/or tables, reviewed drafts of the paper.
- Paola Moscoso performed the experiments, contributed reagents/materials/analysis tools, reviewed drafts of the paper, data Collection.
- Mika Peck conceived and designed the experiments, reviewed drafts of the paper, data Collection.

## Field Study Permissions
The following information was supplied relating to field study approvals (i.e., approving body and any reference numbers):

Ministry for the Environment, Ecuador.

N'' 008-15 -IC-FAU-DNB/MA

## Data Availability
The raw data has been supplied as Supplemental Information.

## Supplemental Information
Supplemental information for this article can be found online at http://dx.doi.org/10.7717/peerj.2108#supplemental-information.

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
