# Peer review of "A new method for ecoacoustics? Toward the extraction and evaluation of ecologically-meaningful soundscape components using sparse coding methods"

_PeerJ, doi:10.7717/peerj.2108_

## Round 0.1 · original submission · Major Revisions

I am very sorry for the delay in getting this manuscript back to you, we had a very difficult time finding referees willing to review this over the Holidays, and have now been waiting on a third referee with acoustic analyses expertise for over a month as they promise to return the review in a "couple of days" each time we remind them. Unfortunately, they have never come through and it would have been very nice to get you some additional feedback on those aspects of your manuscript because it is not my specific area of expertise either. Regardless, rather than delay your manuscript any further, I will make a decision based on the two reviews that I was able to get.

The first referee works in the same area and felt that the more general review of the subject matter was unnecessary, could be greatly reduced or removed entirely and that the focus should instead be shifted to the new methodology. The second referee is not expert in the computational aspects of your work, and so appreciated the review that was provided. It seems to me that a good solution to satisfy both referees would be to move the review sections to a supplement/appendix that includes this information for the interested reader, but still allows you to focus the main text more directly on the subject of the new method. Beyond that, it seems the recommendations of the referees are straight forward and I expect that they should be relatively easy to address, but please feel free to contact me should you encounter any difficulties with the revisions. I look forward to seeing your revised manuscript.

Reviewer 1 ·

Basic reporting

The article has some parts that are not necessary. For instance I suggest to shorten the section 1.1 and 1.2.
The section 1.3 is not necessary, this article is not a review but the presentation of a new methodologies that requires some references but not a presentation of the indices in ecoacoustics.

Experimental design

I suggest to better concentrate on the section 1.4 that is the core of the article that should be integrated with the section 2.3 that should be better described.

Validity of the findings

on line 342 it appears a contradictory statement " A detailed discussion of the methods... " I suggest to discuss in detail the proposed methodology, it's the right place and time.

on line 384 referring to the acoustic niche seems inappropriate

Additional comments

I suggest to concentrate on the proposed methodology reducing the descriptive parts of the other ecoacoustics indices.
I suggest to shorten the section 1.1 and 1.2.
The section 1.3 is not necessary, this article is not a review but the presentation of a new methodologies that requires some references but not a presentation of the indices in ecoacoustics.
I suggest to expand the concept of the niche hypothesis in order to be more clear how the new methodology could contribute to its investigation.
The results seem for the same admission of the authors not clear (line 373-379). These questions should be addressed in the introduction and discussed a the end of the paper. From line 380 to 388, the meaning is not clear for me. I do not understand what is the object of the discussion, if the acoustic niche or the capacity to separate different sound components. This part requires an carefully revision.

·

Basic reporting

The manuscript is very well written, and provides a nice overview of the state of acoustic indices used in the emerging field of soundscape ecology and ecoacoustics. The paper makes a contribution to this literature by applying alternative analyses by which ecologically relevant information could be obtained from acoustic recordings. The limitations of the previously established methods in identifying or comparing spectro-temporal information are well-described, and the potential improvements of the sparse coding methodology are convincingly presented. My comments, from the perspective of an ecologist without computer science background, primarily are suggestions/questions to more effectively communicate the computational science and broad relevance of the methodology to a biology/ecoacoustics audience, to fit with PeerJ scope rather than PeerJ Computer Science.

An overall comment on the basic reporting is that there is no consideration that this study, as well as the literature reviewed in the background/introductory text, currently only considers terrestrial systems, and, in particular, avian communities in forests. The background discussion of “Acoustic Approaches to Biodiversity Assessment” and “Existing Acoustic Indices for Automated Ecoacoustics” pertains to terrestrial habitats, and this should especially be acknowledged in instances where frequency ranges of sound types (e.g. lines 94-95) are being generalized. Given that the stated aims of the type of analysis being presented here (i.e. biodiversity assessment) is relevant to a range of ecosystems, and indeed there are ecologists attempting to use passive acoustic monitoring in a variety of habitats terrestrial and aquatic, a mention of the potential applicability or pitfalls of the sparse coding method for analysis of different soundscape types would enhance the manuscript. Indeed, it would be useful to highlight somewhere that the growing research interest spans systems and approaches are being developed for habitats with very different ecoacoustic characteristics. In the least, it needs to be made clearer to readers that this paper presents the new method within a limited scope of testing a specific terrestrial example.

Experimental design

The research objective and aims of the study are clearly defined, and section 1.4 gives a thorough treatment of the motivations and rationale for the application of sparse coding/ PLCA approaches to the problem of meaningful acoustic monitoring. Still, while many of the examples and descriptions given in this introduction to the approach are extremely useful and fairly accessible, I suggest that some additional explanations and definitions of terms would be useful to effectively present this material within the scope of PeerJ (see line comments below).

Since the Methods and Materials section details a much longer well-replicated data collection than what was used in this “illustrative example”, there needs to be some description about how the three samples were selected for use (lines 272-273), and why more samples could not be used to more rigorously test the new method and potentially garner more information about the behaviour of the acoustic indices compared to the sparse coding method. If this was done, but only three examples are being presented, this should be stated and explained why these examples were selected. Would examining additional samples and investigating the available dataset more thoroughly help to disentangle some of the ecologically relevant questions that are later noted to be inconclusive from this purely illustrative analysis? I’m not familiar with the actual analysis procedure or computational costs, so if the number of examples that could be examined for method validation was limited, it would be appropriate to note that here to justify the selection of only three short recordings from a single day and time.

Validity of the findings

In general, the authors report the significance of findings carefully with full qualification of the limitations in terms of ecological conclusions. I agree that the significance of their approach, as with the other acoustic indices examined, requires far more investigation before strong interpretations can be made.

The example used to evaluate the method is only from a dawn recording and does not contain a lot of the sound sources that may typify many environments and complicate the use of existing acoustic indices (e.g. broadband signals, anthropogenic noise). To better evaluate the benefits of sparse coding SI-PLCA approaches compared to the existing acoustic indices, would it not be most useful to include examples where the other indices are weakest? I agree that the use of “subtle habitat gradients” provided by the dataset has some benefit in testing the ability of the technique to detect subtle differences, however, I would still be most convinced of it’s merits if directly validated with the analysis of samples with broad-spectrum sources (as you noted in lines 351-352, the SI-PLCA2D dictionary appears to be suited for better extracting these elements). Could a comparison of sparse-approximation outputs for example(s) with more or different broadband elements (e.g. cicada or cricket choruses) not be added here?

As an ecologist without computational experience related to the new techniques presented in this manuscript, I am excited to see the development and communication of advanced approaches for improved soundscape analysis. Given that one of the motivations of this manuscript is to highlight a new approach for ecoacoustics, and since the authors rightly indicate that this field is seeking cost-effective and easily interpretable methods, I wonder how difficult (computationally, conceptually) it is for the typical soundscape ecologist to apply this tool to their investigations? Can we use this method for the kinds of long-term high sampling rate datasets that are starting to be collected in passive acoustic monitoring endeavours? In the summary and future work section I am looking for a better idea of how close we are to being able to apply this technique in the ways outlined in the introduction. Are the questions raised in lines 367-369 and 374-379, that are suggested as topics for further work, things we can address in the near-future as a field? How do we actually move forward with this technique if it is to be used as part of the “central and urgent task” of biodiversity assessment? The “Summary and Future Work” section does a good job of reiterating the motivations and promise of the approach presented but is quite vague in offering any idea of whether this method really gets us any closer to meeting the targets stated in lines 3-4. I think this section could be expanded slightly to include some of these more practical and logistical issues. Given that the premise of the manuscript is to lay the foundation for a new methodology for ecoacousticians and conservation scientists, I would find it useful and appropriate to more specifically explain how the authors foresee the application of the sparse coding methods by these user groups.

Additional comments

Specific line comments –

Line 94-95 – this is where I particularly take issue with ignoring the marine soundscape, where the anthropophony, biophony and geophony is associated with different bandwidths. In this section I would suggest making it clear that you are generalizing the acoustic characteristics and reviewing the literature for terrestrial habitats only.

Line 120 – this is also a problem in marine soundscapes that prevents the application of the existing acoustic indices; this is worth mentioning as these indices and associated deficiencies have started to crop up in the marine soundscape literature (e.g. McWilliam & Hawkins 2013, Harris et al. 2015), and acknowledging this would expand the scope of the manuscript if the sparse coding methods presented here could be applicable to marine environments

Line 142 – missing “a” after “presented as…”

Line 166 – dictionary should be defined here rather than in the next paragraph (this is the first usage of what is a very specific meaning of “dictionary”)

Line 170 – “The basic idea is fairly simple” – this sentence is probably needless and vague, and what follows is not necessarily fairly simple to a lot of readers if this is intended for PeerJ rather than PeerJ Computer Science, though it is well explained for a more general biology audience. I suggest removing it or modifying to convey more meaning.

Line 170 – define atomic function

Line 204-205 – what is a “groove-similarity task”?

Line 236 – add “terrestrial” before “habitats”
Line 259 – why is 300m sufficient to avoid “pseudo sampling”? Please provide a reference or explanation for this distance between replicates.

Line 268-270 – Please describe “point counts” more fully as a sampling method. I assume the observer counts all birds within a certain radius, but I am not familiar with the details of this method.

Figure 3 – in legend, indicate which site this is (FP?); what are the x and y variables in the spectrogram approximations? How are these scaled?

Figure 4-6 – axis labels? I would like to at least see the y-axis (frequency) scales on these figures, though Figure 3 makes me unsure of what the scale or variables in the sparse approximation spectrograms are

---

## Round 0.2 · Minor Revisions

I appreciate your careful edits on this version and the revisions have addressed the major concerns of the referees to my satisfaction. The more critical referee felt that the paper was a strange hybrid between a review and experimental work, but if senior folks in the field feel that the critique of current indices is an important aspect of the manuscript, I would hate to stifle that simply on the basis of one referee. From my perspective, this is an opinion of the referee and not a valid scientific criticism of the work, so I am happy to include the sections if you and other experts in the field feel it is worthwhile to do so, and my reading of your response to referee comments suggests to me that you do.

Would you prefer to move forward with this version or reintegrate the section into the introduction? I made a suggestion for what I considered a reasonable path forward, but wanted to let you know that I am open to whatever you feel is best for the manuscript and would like to leave the decision in your hands on whether or not to include that section in the main body of the text.

Thus, I am returning this version for minor revisions to allow you to upload the revised text.

---

## Round 0.3 · accepted · Accept

I am happy to accept your manuscript with the section included as you prefer. I will move this forward into production now.